# A systematic review of the association between perinatal depression and cognitive development in infancy in low and middle-income countries

Matthew Bluett-Duncan[1,2], M. Thomas Kishore[2], Divya M. Patil[3], Veena A. Satyanarayana[2], Helen Sharp[1] *

1 Department of Primary Care and Mental Health, Institute of Population Health, University of Liverpool, United Kingdom, 2 Department of Clinical Psychology, National Institute of Mental Health and Neurosciences, Bengaluru, India, 3 Department of Epidemiology, School of Public Health, University of Washington, Washington, DC, United States of America

* hmsharp@liverpool.ac.uk

**Data Availability Statement:** All relevant data are within the manuscript and its Supporting information files.

## Abstract

The association between perinatal depression and infant cognitive development has been well documented in research based in high-income contexts, but the literature regarding the same relationship in low and middle-income countries (LMICs) is less developed. The aim of this study is to systematically review what is known in this area in order to inform priorities for early intervention and future research in LMICs. The review protocol was pre-registered on Prospero (CRD42018108589) and relevant electronic databases were searched using a consistent set of keywords and 1473 articles were screened against the eligibility criteria. Sixteen articles were included in the review, seven focusing on the antenatal period, eight on the postnatal period, and one which included both. Five out of eight studies found a significant association between antenatal depression (d = .21-.93) and infant cognitive development, while four out of nine studies found a significant association with postnatal depression (d = .17-.47). Although the evidence suggests that LMICs should prioritise antenatal mental health care, many of the studies did not adequately isolate the effects of depression in each period. Furthermore, very few studies explored more complex interactions that may exist between perinatal depression and other relevant factors. More high-quality studies are needed in LMIC settings, driven by current theory, that test main effects and examine moderating or mediating pathways to cognitive development.

## 1. Introduction

The global prevalence of perinatal depression has been estimated at 11.9%, but a recent meta-regression has shown a significant difference between high income countries (HICs) and low- and middle-income countries (LMICs) [1]. According to this review, the mean adjusted pooled prevalence of perinatal depression is 11.4% in HICs and 13.1% in LMICs. These

**Funding:** HS was supported by funding from the Medical Research Council (Ref: MR/N000870/1, https://mrc.ukri.org/) and the Indian Council of Medical Research (Ref: ICMR/MRC-UK/2/M/2015-dcd-1, https://www.icmr.gov.in/). MBD was supported by a Dual PhD Scholarship as part of a collaboration between the University of Liverpool (https://www.liverpool.ac.uk/) and the National Institute of Mental Health and Neurosciences, Bangalore (https://nimhans.ac.in/). The funders had no role in study design, data collection and analysis, decision to publish, or preparation of the manuscript.

**Competing interests:** The authors have declared that no competing interests exist.

findings are supported by another review which reported that the prevalence of common perinatal mental disorders in LMICs is higher than that found in HICs, with weighted mean prevalence reported as 15.9% antenatally and 19.8% postnatally [2]. Both studies drew data from studies that used a combination of diagnostic instruments and symptoms scales.

## 1.1 Sensitive period for cognitive development

Cognitive development refers to the whole range of mental activities and skills, including memory, language, learning, problem solving, perception and social cognition [3]. During infancy cognition is typically assessed in terms of developmental milestones, including motor actions and language skills, and is conceptualised broadly as cognitive development, rather than the specific higher order functions, such as executive function and complex problem solving, that emerge later [4]. In view of this, and in view of the developing nature of the literature in LMIC settings, this review utilises an inclusive and focused conceptualisation of global infant cognitive development that includes early mental, language and psychomotor milestones.

The perinatal period, and beyond into infancy, has received particular attention in HIC research because it represents an important and sensitive stage where children are at their most receptive stage of development [5–7]. From conception, up until around 3 years, the brain exhibits a high level of neural plasticity and rapid synapse formation that enhance the capacity of the child to learn and develop [8]. Neural connections are formed that affect development throughout the life course and provide the basis for future social, emotional and cognitive development [9]. While this is beneficial for development under optimal conditions, adverse life experiences during this time can have long-lasting, detrimental effects [10].

## 1.2 High-income settings

The possible detrimental effects of perinatal depression on infant health and development have been well documented in research from HICs. Significant deleterious impacts have been observed in brain development, regulatory behaviours, acquisition of developmental milestones and various other behavioural, emotional, and physical outcomes [11–13]. In addition, there is a significant body of evidence supporting the presence of an association between perinatal depression and poor infant cognitive outcomes, summarised in a number of systematic reviews and meta-analyses [14–19]. However, although generally significant, effects are generally small and long-term effects tend to be confined to high-risk samples and subgroups of infants experiencing additional risks, suggesting the presence of a complex interaction of different factors, rather than a simple and direct relationship [20]. As such, developmental frameworks have become progressively more complex and HIC research has increasingly turned to the investigation of different mechanisms underlying the transmission of risk from perinatal depression to impaired infant cognitive development [21]. These mechanisms are not the focus of this review, but detailed discussion of the relevant processes can be found elsewhere [9, 22].

## 1.3 Low- and middle-income settings

In LMICs, where infants are generally exposed to a larger number and range of adversities, the impact of factors such as perinatal depression are likely to be magnified [23]. Although depression may act in a similar way regardless of context, the co-occurrence of additional risk factors in LMICs is anticipated to result in increasingly compromised development, relative to that observed in HICs [5]. While a number of broader systematic reviews have included the relationship between perinatal depression and cognitive development in LMIC settings [24–27], there has not yet been a focused synthesis of studies on perinatal depression and cognitive development conducted in the LMIC context. A recent systematic review of studies examining

the impact of perinatal mental health on infant neurodevelopment in LMIC settings [28] included both a wide range of perinatal mental health disorders (e.g. schizophrenia, mania, PTSD, anxiety, depression) which may have very distinct effects on infant development, and a wide range of child outcomes to age 2 (gross and fine motor, cognitive, language, behavior and social-emotional development). The narrative synthesis therefore did not provide an in-depth evaluation of the evidence for effects of perinatal depression per se on cognitive development specifically. The authors concluded "Due to heterogeneity of reported types of maternal health disorders and different domains of developmental outcomes, it was not possible to draw a definitive conclusion about the association between prenatal exposure to maternal mental health and child development" [28, p. 168].

Finally, although there now have been a number of individual studies that have explored the relationship between perinatal depression and infant cognitive development in LMIC settings, the overall literature in this context is not yet well developed and has not been synthesised. As such, it is expected that the majority of research identified in this review will be focused on investigating the presence of main effects of perinatal depression on infant outcomes, rather than on testing the more complex interplay between factors that has more recently emerged in the HIC literature.

### 1.4 Identifying independent effects of depression at different time-points

An important distinction demonstrated in research from HICs is that depression during the antenatal and postnatal periods exert independent effects on development and act on development through different mechanisms [29]. Accordingly, it is important to delineate independent and cumulative effects. In order to establish whether there is an independent effect within these distinct periods, studies need to satisfy two criteria. Firstly, because the presence of a depressive episode in one period is significantly associated with a depressive episode in the other, studies focusing on the influence of one period should adequately control for the effects of the other [30, 31]. Secondly, studies investigating depression in either period on later cognitive outcome should account for the effects of concurrent depressive symptoms at the time of cognitive testing. Perinatal depression is known to increase the likelihood of future episodes of depression so any apparent cognitive impairments may be the result of current, rather than prior perinatal symptoms of depression, reflecting either a true impact on cognition, when assessed by independent observers, or, more often, a view of the child that is distorted by depressive symptoms when cognition is assessed via maternal report [32–34].

### 1.5 Aims of this review

The primary purpose of this review was to synthesise the evidence for the independent and joint effects of antenatal and postnatal depression on the cognitive development of infants aged 0–3 years in LMICs. The strength of the evidence for the independent effects will be discussed in terms of whether studies have adequately controlled for concurrent depressive symptoms and depressive episodes at other time-points in the perinatal period. These findings can then be used to inform recommendations for early intervention and the progression of the literature in this area in terms of how studies in LMICs can be designed to more effectively test current theory.

## 2. Materials and methods

This research adhered to the Preferred Reporting Items for Systematic Reviews and Meta-Analyses (PRISMA) guidelines for systematic review [35]. A protocol was completed prior to the review being carried out and registered on Prospero (CRD42018108589).

## 2.1 Inclusion and exclusion criteria

Observational studies (prospective, cross-sectional, case-control) were eligible for inclusion if they quantitatively assessed the association between maternal depression during pregnancy and/or the first 12 months post-birth and cognitive development in infants aged 0–3 years, and if a) maternal depression was diagnosed through a standardised diagnostic interview, or a validated symptom questionnaire or screening tool, b) cognitive development was assessed using a direct, validated measure of cognitive or language ability, and c) the research was carried out in a LMIC as designated by the World Bank.

Due to the emerging nature of the literature, this review has leaned toward overinclusion of studies, and therefore RCTs were also eligible for inclusion if they a) presented findings on the control arm of the study, b) present findings using baseline data, or c) they appropriately adjusted for intervention effects in the full sample. No historical time-limits were imposed on publication dates. Final searches included studies published in May 2020.

Studies were excluded if a) depressive symptoms were assessed as part of a general mental health assessment from which it was not possible to isolate the effects of depression (e.g. SRQ-20), b) they examined specific patient populations (mother or infant) with any physical disease or disorder that may impact cognitive development, other than maternal depression, c) development was assessed using implied measures of cognition, or d) they were case-series' or case-studies.

## 2.2 Definitions

**2.2.1 Perinatal depression.**   Defined as a non-psychotic depressive episode of mild to major severity that occurs during pregnancy or up to 12 months postnatal [24], diagnosed through standardised diagnostic interviews or validated symptom scales, and not restricted to confirmatory clinical diagnoses.

**2.2.2 Infant cognitive development.**   This review utilises an inclusive conceptualisation of global infant cognitive development that includes early mental, language and psychomotor milestones.

**2.2.3 Low- and middle-income country (LMIC).**   A country designated as a LMIC by the World Bank at the point of data collection.

## 2.3 Search strategy and study selection

Eligible studies were identified using electronic and manual searches in October 2018. Follow-up searches were completed in May 2020 to ensure that all relevant studies were included. PubMed, PsychInfo and CINAHL were identified as the most relevant databases and searched for relevant articles. Keywords relating to the research questions were identified and an overall search strategy was devised using Boolean operators to combine free text and MeSH terms (Fig 1). Free text terms were used uniformly across searches, but minor alterations were made to tailor MeSH terms appropriately to each database. In order to reduce the impact of publication bias, the grey literature was also searched via ProQuest. Finally, reference lists of included articles were hand-searched to identify any relevant articles not revealed by the electronic search.

Study selection was completed in two phases (Fig 2). First, title and abstract screening of all retrieved articles was completed independently by two reviewers (MBD and DP). Studies which were deemed by both reviewers to meet inclusion criteria, or that were unclear, were selected for full-text review. Studies that clearly did not meet inclusion criteria were excluded at this point. Second, a full-text review of the remaining articles was completed independently by both reviewers. Studies that were deemed to meet the inclusion criteria at this point were

| Search #1 | ((Mothers [Text Word] OR Maternal [Text Word] OR Perinatal [Text Word] OR Peripartum [Text Word] OR Prenatal [Text Word] OR Antenatal [Text Word] OR Antepartum [Text Word] OR Pregnancy [Text Word] OR Pregnant [Text Word] OR Trimester [Text Word] OR Postnatal [Text Word] OR Postpartum [Text Word] OR Puerperal [Text Word] OR Puerperium [Text Word] OR post-birth [Text Word])) OR (Peripartum [mh] OR Postpartum period [mh:noexp] OR Prenatal care [mh] OR Postnatal care [mh] OR Postpartum depression [mh]) |
|---|---|
| Search #2 | Depression [Text Word] OR Depressed [Text Word] OR depressive symptoms [Text Word] OR Depression symptoms [Text Word] OR affective disorder [Text Word] OR depressive disorder [Text Word] OR depression [mh] OR postpartum depression [mh] OR depressive disorder [mh] |
| Search #3 | 1 AND 2 |
| Search #4 | Infant [Text Word] OR Infants [Text Word] OR child [Text Word] OR children [Text Word] OR infancy [Text Word] OR baby [Text Word] OR babies [Text Word] OR toddler [Text Word] OR Toddlers [Text Word] OR newborn [Text Word] OR childhood [Text Word] OR pre-school [Text Word] OR infant [mh] OR infant, newborn [mh:noexp] OR child, preschool [mh] |
| Search #5 | Cognition [Text Word] OR cognitive [Text Word] OR language [Text Word] OR IQ [Text Word] OR intelligence [Text Word] OR memory [Text Word] OR perception [Text Word] OR learning [Text Word] OR problem solving [Text Word] OR metacognition [Text Word] OR social cognition [Text Word] OR DQ [Text Word] OR communication [Text Word] OR executive function [Text Word] OR attention [Text Word] OR child development [mh] OR cognition [mh] OR intelligence [mh] |
| Search #6 | 4 AND 5 |
| Search #7 | 3 AND 6 |
| Search #8 | low income population [Text Word] OR low income countr* [Text Word] OR middle income population [Text Word] OR middle income countr* [Text Word] OR low and middle income population [Text Word] OR low and middle income countr* [Text Word] OR developing countr* [Text Word] OR developing nations [Text Word] OR third world [Text Word] poverty [Text Word] OR LMIC [Text Word] OR LAMIC [Text Word] OR africa [Text Word] OR asia [Text Word] OR south america [Text Word] OR central america [Text Word] OR South Asia [Text Word] OR middle east [Text Word] OR poverty [mh] OR Asia [mh] OR Africa [mh] OR south America [mh] OR central America [mh] |
| Search #9 | All countries designated as LMIC by World Bank included here as individual Text Words. |
| Search #10 | 8 OR 9 |
| Search #11 | 7 AND 10 |

**Fig 1. Example search strategy used with PubMed.**

included in the final review. All final decisions were made by consensus between the two reviewers. Where studies remained unclear or there were disagreements between the two reviewers, articles were referred to a third reviewer (HS or TK) who independently reviewed the articles and participated in the group consensus. Reasons for exclusion were recorded.

## 2.4 Quality assessment

Study quality was evaluated independently by two reviewers (MBD and DP) using the Newcastle-Ottawa Scale (NOS) adapted for cohort studies [36]. Studies were appraised in 3 areas: selection, comparability, and outcome. Each of these had a number of criteria which were scored according to NOS guidelines, producing quality ratings of good, fair and poor. Each study was assessed using a standardised form which included instructions and clarifications where necessary for applying the scale to the current subject. One of the criteria from the "selection" section (*"Demonstration that outcome of interest was not present at the start of the study")* was not appropriate to the study design in question since many aspects of cognitive function cannot be assessed early in the perinatal period. However, rather than remove this item and adjust the standardised scoring system, it was decided to simply give all the studies a

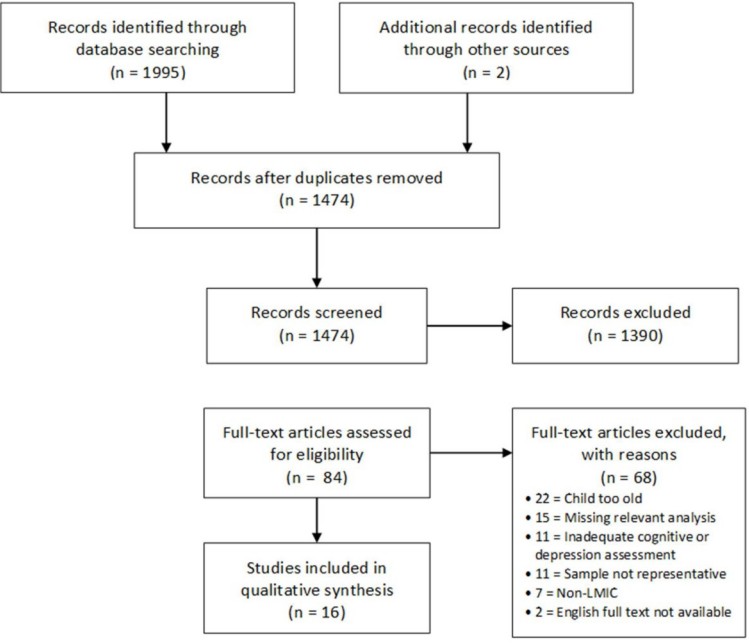

**Fig 2. PRISMA flow diagram.**

score of zero for this item. Thus, it may be that the quality assessments slightly underestimate the quality of each study. Following independent review, any disagreements regarding quality were solved by consensus.

## 2.5 Data extraction

Data extraction was completed independently by two reviewers (MBD and DP) using a standardised data extraction form that was created specifically for this review. Data extracted included country/setting, study design, aims/hypotheses, sample size, sample characteristics, measurement of exposure, measurement of outcome, perinatal stage at exposure, infant age at outcome, statistical approach, confounding variables, mediating or moderating variables, and main results (descriptive, unadjusted, adjusted). Following an initial piloting phase, data were extracted and compared. Any disagreements were resolved by consensus, and any issues that still lacked clarity were referred to HS. Where relevant data were missing study authors were contacted to request specific information.

## 2.6 Analysis

Extraction included the recording of variables that were defined a priori: socioeconomic status, maternal age, maternal education, infant gender, birthweight, intervention (if RCT). If meta-analysis had been possible this set would have represented a minimum set of adjustment variables to compare the effect size of "adequately" controlled studies and "inadequately" controlled studies. In practice, meta-analysis was not possible due to the variation between studies in terms of their context and the measurement tools used. However, this set of variables were considered when examining patterns in the data through narrative synthesis. Articles presenting findings regarding exposure to antenatal and postnatal depression were synthesised and discussed separately.

The adjusted standardised mean difference (d) between depressed and non-depressed group was calculated wherever possible. If this was not possible, unadjusted effect sizes were calculated using group means and standard deviations. The strength of the effect size can be interpreted as follows: 0.2 = small; 0.5 = medium; 0.8 = large [37].

# 3. Results

16 studies were selected for inclusion in this review (Fig 1), 7 with a primary focus on antenatal depression, 8 with a primary focus on postnatal depression, and 1 study that considered both pre- and postnatal exposure. The majority of studies utilised a prospective-cohort design (n = 10). A number of Randomised Controlled Trials (n = 6) were also included if they reported on data from the control arm of the study or they controlled for the effect of the intervention in the final analysis. Finally, there was one study which used a quasi-experimental design.

## 3.1 Approach to narrative synthesis

Findings are grouped by perinatal period according to the main focus of each paper. Significant results are presented first, followed by non-significant findings. Findings are then briefly synthesised in terms of the adherence of each study to the requirements for assessing the independent effects of maternal depression within each period and the inclusion by each study of the a priori key adjustment variables.

Unless otherwise stated, beta coefficients given for main effects are unstandardized and refer to the change in the number of points scored on a particular measure when a mother is depressed, compared to when a mother is non-depressed. Means and standard deviations are reported where available. All depression assessments are maternal self-report unless otherwise stated.

## 3.2 Summary of antenatal studies

Table 1 provides a summary of the key features of the 8 antenatal studies. Antenatal studies were conducted in five different countries: four in South Africa and one in each of China, Mexico, Ukraine and Vietnam. Exposure to maternal depressive symptoms was assessed in four studies by the Edinburgh Postnatal Depression Scale (EPDS) [38], twice with the Beck Depression Inventory (BDI-I and BDI-II) [39, 40], once with the Symptom Checklist-90-Revised (SC-90-R) [41] and once with the Structured Clinical Interview for DSM-IV diagnoses (SCID) [42]. Six studies specified that depression was assessed during the 3rd trimester, one during the 2nd trimester and one simply stated that the assessment was conducted during the antenatal period. The Bayley Scales of Infant Development (BSID) were most commonly utilised for assessing infant outcome, with four studies using the Bayley-III [43] and two studies using the Bayley-II [44]. Another study used the Gesell Scale [45] and the remaining study used the Peabody Picture Vocabulary Test (PPVT) [46] and an executive function battery (EF) [47]. Age at follow-up ranged from 6–30 months. One study was rated as good, six as fair, and one as poor quality, according to the NOS. Adherence of each study to the key adjustment variables defined in section 2.6 are summarised in Table 2.

**3.2.1 Significant findings.** Five out of eight studies found a significant association between antenatal depression and infant cognitive development (Table 1). Four of these studies were fair quality [50–52, 55] and one was poor quality [49]. Results are described below and summarised in S1 Table.

Arguably the most methodologically robust study in this review is Tran et al. [55]. Although the study only received a quality rating of fair, it was the only study to control for both

**Table 1. Key features of included antenatal studies.**

| Authors | Country | Design | Sample (follow up %[a]) | Exposure (Cut-off) | Exposure Timing | Outcome (Subscale) | Outcome Timing | Study Quality | Significant Association Found? |
|---|---|---|---|---|---|---|---|---|---|
| Bandoli et al. [48] | Ukraine | RCT | 754 (45.6) | BDI ($\geq$10) | 32 weeks gestation | Bayley-II (MDI) | 6 & 12m | Fair | No |
| Breen et al. [49] | South Africa | Prosp. | 149 (77.2) | BDI-II ($\geq$15) | 28–32 weeks gestation | Bayley-III (Cognitive; Language) | 24m | Poor | Yes |
| Donald et al. [50] | South Africa | Prosp. | 1143 (73.0) | EPDS (>13) | 28–32 weeks gestation | Bayley-III (Cognitive; Language) | 24m | Fair | Yes |
| Lin et al. [51] | China | Prosp. | 398 (56.5) | SCL-90-R (N/A) | 28–36 weeks gestation | Gesell Scale (Language) | 24-30m | Fair | Yes |
| Munoz-Rocha et al. [52] | Mexico | Prosp. | 760 (62.2) | EPDS ($\geq$13) | 3rd Trimester | Bayley-III (Cognitive; Language) | 24-30m | Fair | Yes |
| Murray et al. [53] | South Africa | RCT | 449 (58.5) | SCID (N/A) | Antenatal period | Bayley-II (MDI) | 18m | Good | No |
| Rotheram-Fuller et al. [54] | South Africa | RCT | 1238 (80.0) | EPDS ($\geq$13) | 26 weeks gestation. | Executive Function Battery (OS, SS, STS) | 36m | Fair | No |
| | | | | | | PPVT | 36m | | |
| Tran et al. [55] | Vietnam | Prosp. | 497 (76.1) | EPDS-Vietnam ($\geq$4) | >28 weeks gestation. | Bayley-III (Cognitive) | 6m | Fair | Yes |

Key: Prosp. = Prospective; BDI = Beck Depression Inventory; EPDS = Edinburgh Postnatal Depression Scale; SCL = Symptom Checklist; SCID = Structured Clinical Interview for DSM-IV; PPVT = Peabody Picture Vocabulary Test; MDI = Mental Development Index; OS = Operation Span; SS = Silly Sounds; STS = Something's the Same.

[a] Follow-up was calculated from the information provided regarding the number of mother-child dyads who completed developmental assessments and/or were included in final analyses.

postnatal and concurrent depression. Using a prospective design to assess a sample in Vietnam, this study found a small effect of depression in that infants of depressed mothers (M = 97.9, S.D. = 14.1) scored lower than infants of non-depressed mothers (M = 100.0, S.D. = 12.8, $d$ = 0.15) on the Bayley-III. Importantly, the authors demonstrated that a significant association between depression and cognitive development at 6 months remained after controlling for both postpartum and concurrent depressive symptoms (B = -4.80, p < 0.05). Lin et al. [51] also controlled for concurrent depression but not postnatal depression. The authors assessed a relatively high SES sample in China using a prospective design. Depression was associated with significant deficits in language scores on the Gesell Scale at 24–30 months (B = -13.18, p = 0.01). Unfortunately, no cognitive composite score was reported which means that the findings are not easily comparable with other studies.

**Table 2. Key *a priori* adjustment variables identified for antenatal studies.**

| Study | PND | Concurrent Depression | SES | Maternal Age | Maternal Education | Infant Gender | Birthweight |
|---|---|---|---|---|---|---|---|
| Bandoli et al. [48] | No | No | **Yes** | **Yes** | **Yes** | No | No |
| Breen et al. [49] | No | No | No | No | No | No | No |
| Donald et al. [50] | No | No | **Yes** | **Yes** | **Yes** | **Yes** | **Yes** |
| Lin et al. [51] | No | **Yes** | **Yes** | **Yes** | **Yes** | **Yes** | No |
| Munoz-Rocha et al. [52] | No | No | No | **Yes** | **Yes** | **Yes** | **Yes** |
| Murray et al. [53] | **Yes** | No | **Yes** | **Yes** | **Yes** | No | No |
| Rotheram-Fuller et al. [54] | No | No | **Yes** | No | **Yes** | No | No |
| Tran et al. [55] | **Yes** | **Yes** | **Yes** | **Yes** | **Yes** | No | **Yes** |

The three other studies that found a significant association between antenatal depression and cognitive outcomes did not account for postnatal or concurrent depression in their analyses and so their results need to be interpreted with caution. Outside of this consideration, however, all are robust studies and are discussed next.

Two studies utilised data from the same prospective sample, the Drakenstein Health Study in South Africa. Breen et al. [49] used a nested sub-sample while Donald et al. [50] used data from the whole sample. Interestingly, the two studies used different tools to assess antenatal depression, with Breen et al. using the BDI-II and Donald et al. using the EPDS. Breen et al. [49] found that, at 24 months, infants of mothers who were depressed scored significantly lower on the cognitive subscale of the Bayley-III (M = 83.7, S.D. = 6.2) than infants of non-depressed mothers (M = 90.0, S.D = 7.8, $d$ = 0.93) and they also scored significantly lower on the language subscale (M = 83.8, S.D. = 14.1) than those of the non-depressed group (M = 87.1, S.D. = 9.9, $d$ = 0.29). In the full Drakenstein Health Study sample, Donald et al. [50] reported unadjusted and adjusted associations between antenatal depression and both cognitive and language outcomes, also using the Bayley-III. Unadjusted bivariate linear regression analyses were carried out on the raw scores of the cognitive, receptive language and expressive language subscales of the Bayley-III for the full sample and after splitting the sample by gender. They reported a significant association between antenatal depression and cognitive outcomes in boys only (B = -1.58, $p <$ 0.05), and expressive language for the full sample (B = -1.06, $p <$ 0.05) and girls only (B = -1.96, $p <$ 0.05). Adjusted analyses revealed a significant association between antenatal depression and cognitive outcomes at 24 months for the full sample after adjusting for a comprehensive set of confounders ($\beta$ = −1.03, $p$ = 0.027, $d$ = 0.21). Final results for the language subscale were not reported.

Similarly, Munoz-Rocha et al. [52] found evidence for a significant association between antenatal depression and both cognitive and language development at 24 months in a Mexican cohort study. This study was designed to investigate the role of blood manganese at different points in the development of the foetus and incorporated this into their final models. In the first model, which included 3$^{rd}$ trimester blood manganese as a covariate, depression during the 3$^{rd}$ trimester was associated with significant deficits in both cognitive (B = -2.40, $p <$ 0.01) and language skills (B = -2.47, $p <$ 0.01) assessed by the Bayley-III. However, in the second model, which excluded antenatal levels of blood manganese but included levels of cord blood manganese during birth, the coefficient was of similar magnitude but was only marginally significant for both cognitive (B = -2.20, $p$ = 0.06) and language skills (B = -2.17, $p$ = 0.08).

**3.2.2 Non-significant findings.** The three studies that did not find a significant association between antenatal depression and cognitive outcomes received good [53] and fair [48, 54] quality ratings.

The key difference between Murray et al. [53] and many of the others in this review is that exposure to antenatal depression was confirmed via diagnostic interview (SCID). In an RCT from South Africa the authors first report an unadjusted ANOVA showing a significant difference between Bayley-II mental development index scores for infants of depressed (M = 81.4, S.D. = 10.1) and non-depressed mothers (M = 84.8, S.D. = 5.1) (F(1,262) = 4.4, p = 0.04, d = 0.33). This moderate effect remained significant after adjusting for 2 month ($p <$ 0.03) and 6 month ($p <$ 0.05) postnatal depression. However, the effect size was attenuated and became marginal once intervention, risk and the intervention by risk interaction term were included in the final model (F = 3.1, $p$ = 0.08, $d$ = 0.27). Risk was a composite measure that included items relating to maternal age, maternal education and SES. Although non-significant, these findings are indicative of a role for antenatal depression in infant cognitive development.

Rotheram-Fuller et al. [54] utilised data from an RCT in South Africa that was investigating the efficacy of a home visiting intervention carried out by community health workers. The

sample was split into four groups relating to the period during which depression was present: never depressed, antenatal, postnatal, antenatal/postnatal. Although this approach does help to isolate the effects of antenatal depression from exposure to postnatal and chronic depression, the postnatal depression group was created from assessments at 2 weeks and 6, 18 and 36 months, which goes beyond the cut-off of 12 months defined as the postnatal period in the inclusion criteria for this review. There were no significant differences between groups at 36 months on the PPVT or on two of the EF battery tasks. This study did not control for the intervention but have reported that there was no effect of the intervention, independently or in combination with maternal depressive group, on cognitive outcome. Bonferroni's method for computing p values and confidence intervals was used to account for multiple comparisons.

Finally, Bandoli et al. [48] found no evidence of an association with antenatal depression in an RCT based in Ukraine. It is important to note that the authors were also interested in the effect of periconceptional alcohol use and so the sample included a higher proportion of moderate to heavy drinkers than would normally be found in a general population. Subsequently, final models included periconceptional alcohol use as a covariate. The authors reported a non-significant association between exposure to depressive symptoms at 32 weeks gestation and Bayley-II mental development index scores at 6 months (B = -1.96, p > 0.05) and at 12 months (B = -0.16, p > 0.05). There was also no significant mean difference between groups when restricted to mothers who had not used alcohol at or around conception at either 6 or 12 months.

### 3.3 Summary of postnatal studies

Table 3 provides a summary of the key features of the postnatal studies. Postnatal studies were conducted in 8 different countries: two in Bangladesh, and one in each of Barbados, Brazil, India, Pakistan, South Africa, Uganda and Vietnam. Exposure to maternal depressive symptoms was assessed using the EPDS in three studies, the Centre for Epidemiologic Studies Depression Scale (CES-D) [56] in two studies, and the Zung Depression and Anxiety Scale (ZDAS) [57], the Aga Khan University Anxiety and Depression Scale (AKUADS) [58] and the Mini International Neuropsychiatric Interview (MINI) [59] in one study each. Infant outcome was assessed with a wide range of tools. The Bayley-II was utilised in three studies and the Bayley-III in one study. The Mullen Scales of Early Learning (MSEL) [60], the Griffiths Mental Development Scale [61], the Early Childhood Development Tool (ECD), and the Developmental Assessment Scales for Indian Infants (DASII) [62], and the executive function battery were all used once. Exposure to depression was assessed from 2 months to 12 months and age at follow up ranged for 3 months to 36 months. Several studies assessed the child at more than one time-point or with more than one tool. Adherence of each study to the key adjustment variables defined in section 2.6 is summarised in Table 4.

**3.3.1 Significant findings.** Out of nine studies reviewed, four found an association between postnatal depression and cognitive outcomes (Table 3). One was good quality [69], two were fair quality [65, 68] and one was poor quality [58]. Results are described below and summarised in S2 Table.

Quevedo et al. [69] was the only postnatal study to receive a 'good' quality rating. Two characteristics set this study apart from many of the others. First, the authors took concurrent depression into account, and secondly, the authors used a diagnostic interview to assess maternal depression. Infants were assessed for language development at 12 months using the Bayley-III and split into 4 groups according to mothers' depression. Group means and unadjusted effect size, with the no depression group used as the reference group, are reported in parentheses. The four groups were as follows, none (M = 108.6, S.D. = 17.0), postpartum (M = 107.2, S.

**Table 3. Key features of included postnatal studies.**

| Authors (year) | Country | Design | Sample (follow-up %[a]) | Exposure (cut-off) | Exposure Timing | Outcome (Subscale) | Outcome Timing | Study Quality | Significant Association Found |
|---|---|---|---|---|---|---|---|---|---|
| Ali et al. [58] | Pakistan | Quasi-Exp. | 420 (91.6) | AKUADS ($\geq$17) | 1, 2, 6, 12, 18, 24 & 36m | ECD Tool (Cognitive; Language) | 1-12m, 18, 24, 30 & 36m | Poor | Yes |
| Black et al. [63] | Bangladesh | RCT | 346 (63.9) | CES-D (N/A) | 12m | Bayley-II (MDI) | 6 & 12m | Fair | No |
| Familiar et al. [64] | Uganda | Prosp. | 228 (95.6) | CES-D ($\geq$16) | 6m | MSEL (Cognitive) | 6 & 12m | Poor | No |
| Galler et al. [64] | Barbados | Prosp. | 226 (48.5) | ZDAS ($\geq$50) | 7w & 6m | Griffiths Scale (DQ) | 7w, 3 & 6m | Fair | Yes |
| Garman et al. [65] | South Africa | RCT-Control | 594 (58.2) | EPDS (N/A) | 2w, 6 & 18m | Bayley-II (MDI) | 18m | Fair | No |
| | | | | | | Executive Function Battery (OS, SS, STS) | 36m | | |
| Hamadani et al., [66] | Bangladesh | Prosp. | 488 (93.7) | EPDS ($\geq$10) | 6w & 6m | Bayley-II (MDI) | 12m | Fair | No |
| Patel et al. [67] | India | Prosp. | 171 (52.0) | EPDS ($\geq$11) | 6w | DASII (MDI) | 6m | Fair | Yes |
| Quevedo et al. [68] | Brazil | Prosp. | 342 (86.5) | MINI (N/A) | 1-2m | Bayley-III (Language) | 12m | Good | Yes |
| Tran et al. [55] | Vietnam | Prosp. | 497 (76.1) | EPDS-Vietnam ($\geq$4) | 8w & 6m | Bayley-III (Cognitive) | 6m | Fair | No |

Note: Prosp. = Prospective; AKUADS = Aga Khan University Anxiety and Depression Scale; CES-D = Centre for Epidemiological Studies—Depression; ZDAS = Zung Depression and Anxiety Scales; EPDS = Edinburgh Postnatal Depression Scale; MINI = Mini International Neuropsychiatric Interview; ECD Tool = Early Childhood Development Tool; MSEL = Mullen Scales of Early Learning; DASII = Developmental Assessment Scale for Indian Infants; MDI = Mental Development Index; DQ = Developmental Quotient; OS = Operation Span; SS = Silly Sounds; STS = Something's the Same.

[a] Follow-up was calculated from the information provided regarding the number of mother-child dyads who completed developmental assessments and/or were included in final analyses.

D. = 16.5, $d = 0.07$), current (M = 105.9, S.D = 14.1, $d = 0.16$) and postpartum/current (97.4, S.D. = 15.4, $d = 0.66$). Adjusted analyses showed that infants of mothers who reported depression had significantly lower language scores (B = -2.87 $p < 0.01$, $d = 0.17$) and post-hoc tests revealed that this effect was only significant for the combined postpartum and current depression group.

**Table 4. Key *a priori* adjustment variables identified for postnatal studies.**

| Study | AND | Concurrent | SES | Maternal Age | Maternal Education | Infant Gender | Birthweight |
|---|---|---|---|---|---|---|---|
| Ali et al. [58] | No | N/A | No | **Yes** | **Yes** | No | No |
| Black et al. [63] | No | N/A | **Yes** | Yes | **Yes** | **Yes** | No |
| Familiar et al. [64] | No | No | **Yes** | Yes | **Yes** | **Yes** | No |
| Galler et al. [65] | No | No | **Yes** | Yes | **Yes** | No | **Yes** |
| Garman et al. [66] | No | No | **Yes** | Yes | **Yes** | No | No |
| Hamadani et al., [67] | No | No | No | No | No | No | No |
| Patel et al. [68] | No | No | No | No | **Yes** | No | **Yes** |
| Quevedo et al. [69] | No | **Yes** | **Yes** | No | **Yes** | **Yes** | **Yes** |
| Tran et al. [55] | **Yes** | **Yes** | **Yes** | **Yes** | **Yes** | No | **Yes** |

All other studies with significant findings did not account for either antenatal or concurrent depressive symptoms. Patel et al. [68] used a case-control design nested within a prospective cohort to assess mother-infant dyads in India. When DASII scores were treated as continuous, there was a significant difference between the mean cognitive scores of the depressed (M = 86.4, S.D = 8.33) and the non-depressed (M = 90.3, S.D. = 9.27, $d$ = 0.44) groups. When treated as dichotomous, with scores <85 considered to be at risk for developmental delay, infants of postnatally depressed mothers were significantly more likely to be delayed than infants of non-depressed mothers (OR = 3.3, p = 0.02). Galler et al. [65] assessed cognitive outcomes using the Griffiths Mental Development Scale in a prospective cohort from Barbados. Following a repeated measures analysis of cognitive outcomes at 7 weeks, 3 months and 6 months, the authors report a significant association between depression at 7 weeks and cognitive outcomes at 3 months, after adjusting for confounders (F(3, 78) = 2.09; p < 0.02). Further, although not statistically significant, a similar linear relationship was also present between 7-week depression and 6 months cognitive outcomes.

Finally, Ali et al. [58] utilised a quasi-experimental approach to assess developmental outcomes in Pakistan. Depression was assessed at 1, 2, 6, 12, 18, 24 and 36 months, while cognitive and language outcomes were assessed using the ECD every month in the first postnatal year and then at 18, 24, 30 and 36 months. The authors first looked for an interaction between PND and infant age on development and followed up on any significant interactions with a cross-sectional analysis of the impact of PND at 2, 6, and 12 months. Infants of depressed mothers were significantly more likely to be delayed at the 6th month of follow-up (OR = 3.3 (95% CI: 1.1, 9.9)) and the 12th month of follow-up (OR = 6.8 (95% CI: 3.0, 15.7)). There was no significant association between depression and language.

**3.3.2 Non-significant findings.** Five out of nine postnatal studies did not find a significant association between postnatal depression and cognitive outcomes. Of these five studies, four were fair quality [55, 63, 66, 67] and one was poor quality [64].

Tran et al. [55] was the only postnatal study to control for the effects of both antenatal and concurrent depressive symptoms. However, unlike the antenatal findings, the authors did not find a significant association between postnatal depression and Bayley-III cognitive scores at 6 months (B = 1.26, p > 0.05). Interestingly, the unadjusted means indicated that infants of depressed mothers (M = 102.9, S.D. = 14.1) actually scored higher than infants of non-depressed mothers (M = 98.5, S.D. = 13.1, $d$ = 0.32).

Hamadani et al. [67] found no significant difference in Bayley-II mental development index scores between infants of mothers depressed at 6 weeks and/or 6 months (M = 99.7, S.D. = 10.8) and those reporting no depressive symptoms (M = 100.6, S.D = 12.1, $d$ = 0.08) in a sample from Bangladesh. Familiar et al. [64] assessed cognitive outcomes at 6 and 12 months in a Ugandan sample. Using a repeated measures approach, the authors found no association between depression and the composite cognitive score (B = -2.39, $p$ = 0.15).

Garman et al. [66] utilised data from the same South African RCT as Rotheram-Fuller et al. [54] but drew data from the control arm of the study. Latent growth analysis was used to assess the effects of distinct trajectories of maternal depression. Mothers' depression was characterised as chronic low, early postpartum (6m), late postpartum (18m) and chronic high (6 & 18m). The authors report that there was no significant difference between the early postpartum depression (M = 10.2, S.D. = 2.3) and the chronic low group (M = 10.1, S.D = 3.0, $d$ = 0.08) on Bayley-II mental development index scores at 18 months (β = 0.08 p = 0.905). There were also no significant associations between early postpartum depression and performance on any of the tasks that formed the EF battery at 36 months (B's = -0.75, -0.29, -0.11, all $p$'s > 0.05). Similarly, there were no significant effects for any of the other depression groups. It is unclear whether beta coefficients reported are standardised or not.

Finally, Black et al. [63] used a cross-sectional design to assess mother-infant dyads in Bangladesh who were participating in an RCT. Depressive symptoms and cognitive outcomes were assessed at 12 months using the CES-D and the Bayley-II, respectively. The unadjusted association between depression and cognitive outcome was significant (r = -0.14, $p < 0.05$), but the adjusted model, including earlier cognitive development at 6 months, revealed a non-significant association (B = 0.09, p > 0.05). Although an unadjusted association is reported, this may reflect the effects of the micronutrients given within the RCT.

## 4. Discussion

This systematic review of 16 studies investigating the association between maternal perinatal depression and infant cognitive outcomes in LMICs provides relatively strong evidence for an effect of antenatal depression but more inconsistent findings amongst postnatal studies. Five out of eight antenatal studies found a prospective negative association between depression and cognitive outcomes. However, only four out of nine postnatal studies found evidence of a prospective or cross-sectional negative association with infant cognitive development. While limited in number and mixed in design, most studies were of fair to good quality, with only three studies being rated as poor. It is important to consider and interpret this pattern of findings in the context of key study design features, as well as the size and quality of the evidence base, and the potential moderated effects that were not tested. This will be discussed in the following sections.

### 4.1 Is the perinatal period a sensitive period for cognitive development?

**4.1.1 Antenatal depression.** There is fairly consistent evidence of a significant association between antenatal depression and infant cognitive development. All five antenatal studies that used a prospective, observational design report a significant association ($d = 0.21–0.93$) between antenatal depression and infant cognitive outcomes [49–52, 55]. A quality rating of fair was given to four of these studies, and poor to one. Although, these findings echo the conclusions of Van Den Bergh et al. [19] that exposure to various types of stress in utero, including depression, can have lasting effects on offspring development, the studies reviewed here actually find a more consistent effect of antenatal depression than studies in HICs, where there is a broadly even split between those finding a significant effect on cognitive development [70–75] and those that do not [76–80]. It is possible that these differences arise because antenatal depression is experienced alongside other risk factors more prevalent in LMIC settings, such as intimate partner violence or poor nutrition, that may exacerbate foetal programming effects [25].

Interestingly, the evidence suggests that the effects of antenatal depression are relatively long-term. Four antenatal studies found an association between depression during the 3rd trimester and cognitive development at 24–30 months. This is an important finding because it is between the 2nd and 3rd year that a child's cognitive ability begins to stabilise and form the foundation for future development [7, 81]. Thus, any impairment at this age is likely to have a lasting effect on child outcomes. This is discussed further in section 4.4.

Concerning the criteria for determining evidence of an independent effect of antenatal depression, adherence was generally low in the reviewed studies. One study controlled for postnatal depression only [53], one controlled for concurrent depression only [51], and one controlled for both postnatal and concurrent depressive symptoms [55]. One additional study isolated the effects of antenatal depression by splitting the sample into groups based on the timing of exposure [54]. Although representing only a small proportion of the reviewed papers, it is worth noting that in all but one of these studies, the effect of antenatal depression

remained significant even after controlling for postnatal or concurrent depression (although the main effect found in Murray et al. [53] became marginal after including additional factors in the model).

Of particular note are the findings of Tran et al. [55] who used path analysis to simultaneously test the effects antenatal, postnatal and concurrent depression, and report that only antenatal depression was a significant predictor of cognitive development. This suggests that the impact of antenatal depression on cognitive outcomes cannot be accounted for by postnatal or concurrent depressive symptoms. This study also controlled for other key factors such as maternal education, SES and infant birthweight, all of which have been identified as important factors in a child's cognitive development [50]. Another key study is Donald et al. [50]. Although they did not control for postnatal or concurrent depression, they did control for each of the key adjustment variables that were identified a priori. By doing this, the authors have ruled out confounding effects from relevant third variables, thereby providing a good degree of confidence that the reported effect was in fact due to antenatal depression. It is important that these findings are replicated in equally well-designed studies.

All of the studies which did not find a significant association between antenatal depression and cognitive outcomes were RCTs and this may have influenced results. In fact, this is clearly the case with Murray et al. [53], where the initial significant effect of antenatal depression was attenuated (p = .06) after the inclusion of a risk by intervention interaction term in the model. Although neither predicted cognition independently, exploration of the interaction between the two factors showed that the intervention improved cognitive performance in low-risk infants. This suggests that merely controlling for the effects of the intervention in an RCT may not adequately account for more complex interactive effects with other factors.

In the case of Bandoli et al. [48], it is pertinent to consider the nature of the sample, which consisted of an index group who reported moderate to heavy drinking during the periconceptional period, and a control group with low or no exposure to alcohol at that point. It is possible that this influenced the results of the study, although there was no mean difference in cognitive scores for infants of depressed or non-depressed mothers within the control group either. Another consideration is that this study took place in Ukraine. Although categorised as a LMIC by the World Bank, Eastern European culture is very different from African or Asian culture, where most of the studies in review are drawn from, making direct comparisons with other studies more difficult.

Finally, while Rotheram-Fuller et al. [54] were able to isolate the effects of antenatal depression, they did not actually control for the effects of the intervention. Although the authors investigated the effects of the intervention on cognitive development and found none, either independently or in interaction with maternal depressive status, it is possible that the intervention may have had some undetected influence on results.

**4.1.2 Postnatal depression.** Evidence for an independent postnatal effect is far less consistent. Three studies, all rated as fair quality, found a significant and negative prospective association (d = .17 -.47; OR = 3.3–6.8) between postnatal depression and cognitive outcomes [66, 68, 69], and one study, rated as poor, found a cross-sectional association [58]. The remaining five postnatal studies, rated poor to fair quality, failed to find any direct association between maternal depressive symptoms and cognitive development, although three of these were RCT designs.

Tran et al. [55] was the only study to control for antenatal and concurrent depressive symptoms and they did not find a significant association between postnatal depression and cognitive outcomes. Another study found that postnatal symptoms at 2 months and concurrent symptoms at 12 months did not independently predict language outcomes, but that there was an association found for chronic exposure to 2-month and 12-month depression [69]. This

finding illustrates the importance of investigating the effects of concurrent depressive symptoms and corresponds to findings from HIC literature that concurrent depressive symptoms may account for the perceived effect of postnatal depression on cognitive outcomes [82].

Although a recent meta-analysis found a significant but small effect of postnatal depression in HICs [18] closer inspection of individual studies suggests that the effect is generally more apparent and pronounced in high-risk samples [83–86]. Thus, due to the higher levels of adversity typically present in LMIC settings, it was expected that the effect of postnatal depression would actually be larger, or at least more consistently found. Somewhat surprisingly, this was not the case. Rather, findings regarding the influence of postnatal depression on cognitive outcomes in LMICs appear to be characterised by the same inconsistencies as those from HICs. Although many of the studies controlled for variations in a variety of risk indicators within each sample, it was still expected that the overall level of risk relative to that found in a HIC would exacerbate any influence of postnatal depression. While it is not clear why this was not the case, it is plausible that the interaction between individual socio-economic risk factors and maternal depression functions differently in LMIC settings [66]. It is also possible the distinct profile of risk in LMIC settings, particularly the severity and co-occurrence of risk factors [23, 25], may be exerting such a strong independent effect on infant cognition that the unique influence of postnatal depression is not sufficient to predict differences in cognitive development.

It should also be noted that HIC studies have demonstrated that chronic depressive symptoms throughout infancy can have a greater impact on a number of domains infant development, including cognition, than a single postnatal episode in a variety of high- and low-risk cohorts [87–90]. In the current review, only two studies examined chronic exposure in the postnatal period. While Garman et al. [66] did not find an effect of chronic depression on development, Quevedo et al. [68] found that only exposure to chronic depression predicted poorer language outcomes. This paucity and inconsistency highlights chronicity of exposure as an important area of research for future studies in LMIC settings.

**4.1.3 Moderated effects.** Although the main purpose of this review was to synthesise evidence for a direct association between perinatal depression and cognitive outcomes there are a few studies which investigated moderated associations and are worth mentioning briefly. Bandoli et al. [48] found no direct association but did find that female infants exposed to both peri-conceptional alcohol use and antenatal depression performed significantly worse than males exposed to the same. Black et al. [63] also did not find a direct association between postnatal depression and change in cognitive skills from 6–12 months. However, the interaction between depression and infant temperament was significant, with irritable infants of depressed mothers showing cognitive impairments, while easy-going infants did not. These results suggest that children with irritable temperaments may be at additional risk for poor development when their mothers are depressed. They also found that caregiving was significantly and negatively associated with maternal depression, and that it mediated the effect of the depression-temperament interaction on cognitive outcomes. This is consistent with HIC literature which has highlighted the dynamic interaction between maternal mental health, caregiving and infant temperament [22, 91]. Finally, Ali et al. [58] found a significant interaction between father's income and depression, with infants of depressed mothers whose fathers earned less than 3500 rupees per month significantly more likely to show language delay. This suggests that differing levels of socioeconomic risk within LMIC settings play a significant role in determining how detrimental perinatal depression is to infant development. Alternatively, paternal income may be an index of paternal education, with lower incomes reflecting lower levels of education. Parental education may be linked to language development through its impact on the level of cognitive stimulation in the home or more directly via heritability effects [7].

## 4.2 Methodological considerations and limitations of reviewed studies

**4.2.1 Perinatal depression assessment.** In this review, all but three studies used screening tools to assess maternal depression and this may have contributed to a lack of consistency in significant findings. This is because symptom questionnaires are more likely to also detect mild or moderate forms of depression, which may lead to an overall underestimation of effect of depression on development [16]. Clinical diagnostic tools focus on symptoms at a level that meet diagnostic threshold and are therefore more likely to represent significant impairment [29]. Unfortunately, there were an insufficient number of studies in this review that used a diagnostic interview to draw any conclusions as to whether infants of mothers with a clinical diagnosis of depression were any worse off than infants of mothers who were assessed using a screening instrument.

**4.2.2 Cognitive development assessment.** Most studies included in the review used measures which generated a standardised composite of cognitive functioning as the outcome. However, even between different versions of the same measure there may be stark differences in the contributing items. For example, the Bayley-II composite includes items relating to language, while Bayley-III has separate cognitive and language subscales. Three antenatal studies and one postnatal study examined both cognitive and language development separately using distinct subscales, while one study in each period looked exclusively at language development. Two studies employed specific measures of executive function rather than broader indices such as composite scores.

One pattern to emerge was the difference in results between studies using the $2^{nd}$ and $3^{rd}$ editions of the Bayley Scales. Across the whole perinatal period the 5 studies which used Bayley-III found a significant effect of maternal depression on cognitive outcomes, while all 5 of the studies which used the older Bayley-II did not. Interestingly, one of the primary justifications detailed for the changes made in Bayley-III was the removal of certain items that may be biased in favour of certain racial or ethnic groups [92]. It is possible that these adaptations enhanced the cross-cultural validity of the instrument, leading to a greater sensitivity to changes in development in LMIC settings. This idea is further supported by the observation that the two studies that used cognitive assessments which had been developed specifically for the culture in which they were being used both report significant results [58, 68].

**4.2.3 Timing of exposure.** Six out of the eight antenatal studies assessed exposure during the $3^{rd}$ trimester, one did not clarify the specific time of assessment, and the other assessed exposure during the $2^{nd}$ trimester. While the fact that five of the $3^{rd}$ trimester studies found a significant association does suggest that the $3^{rd}$ trimester is a sensitive period for development, the lack of assessments at other points of pregnancy precludes conclusions about timing effects. More work is required in this area to investigate whether the impact of antenatal depression varies according when it occurs during foetal development.

There was no distinct pattern of results with regards to the timing of exposure in postnatal studies. Exposure was assessed at various points across the first 12 months and there is too much variability to conclude that any particular period within this timeframe significantly influences the effects of postnatal depression.

**4.2.4 Confounding variables.** There were no clear patterns that emerged in the data with regards to the key adjustment variables, with the exception that studies that controlled for more variables tended to report smaller effects. Interestingly, although not one of the key variables identified, only one study controlled for the effect of nutrition in their analyses. Nutrition is thought to play an important role in cognitive development and so studies in the future should seek to include it where possible. One of the major difficulties in comparing the findings of the different studies in this review is the wide array of adjustment variables included by

different authors. The studies reviewed here varied significantly both in the number and nature of the variables that were controlled for in analyses. Even where the same variables were included, they varied in how they were measured or conceptualised. It is possible that these variations contributed to the inconsistency observed in the overall findings. Thus, a key recommendation from this synthesis is that it is important to control for a wide array of potentially confounding variables, but that researchers should strive to ensure consistency between studies in terms of the confounders included and how they are measured.

## 4.3 Review limitations

The main limitation of this review is the relatively small number of studies that were eligible for inclusion. The main reason for this is that perinatal mental health is very much an emerging field in many LMICs. As a result, one of the recurring themes of this review has been the statement that more studies are required in certain areas in order to draw firm conclusions. Bearing this in mind, the eligibility criteria for inclusion in the review were less narrow than may have been applied in a more developed field. One way in which the reach of the review was extended was to include RCTs under specific conditions. Although every effort was made to ensure that only findings that had not been confounded by the effects of any intervention were included in the final synthesis, RCT interventions may have had an undetected effect on cognitive development, thereby confounding results and potentially adding to the inconsistent findings. When only considering prospective cohort studies, all five antenatal studies and three out of six postnatal studies found a significant, negative direct association between maternal depression and cognitive outcomes. Although postnatal findings remain inconsistent when this restriction is applied, the case for an antenatal effect becomes contextually stronger.

Another potentially limiting factor is the broad categorisation of LMIC countries. While there are clear economic similarities within the countries designated as low- and middle-income included in this study, there are also very distinct cultural differences between them, which could influence the effect of perinatal depression on cognitive development. However, the limited number of studies mean that it would not have been feasible to produce a systematic review for each country. Instead, while the heterogeneity amongst studies is not ideal, this review was able to draw together a previously disparate collection of studies and synthesise what is known and what is lacking in the existing literature. Similarly, cognitive outcomes are considered across a relatively heterogenous age ranges throughout infancy. Although it is possible that this may have affected results, there were no clear patterns in the findings to indicate markedly different effects at distinct ages. Additionally, the developmental period that spans infancy is often considered as a uniform period of accelerated development [5].

Although the risk of reporting bias is an ever-present in systematic reviews, the search strategy for this review included a systematic search of ProQuest for grey literature in the form of unpublished conference abstracts and dissertations. Additionally, the Newcastle-Ottowa Scale for Cohort Studies was used to assess study quality in a number of domains, including participant selection and recruitment, sample attrition, and the methods and measures of exposure and outcome assessment. Key features identified in each study are given in Tables 1 and 3, and all but three studies received a quality rating of fair or good. A further limitation is that not all studies published mean scores for the depressed and non-depressed groups in their sample, and while some authors responded to requests for this information, others did not. This meant that even the most rudimentary comparison of unadjusted effect sizes could not be carried out. In the future, authors should ensure they present this basic descriptive information to aid comparison with other studies.

## 4.4 Implications for clinical practice and future research

The results of this review suggest that maternal depression during pregnancy can have a significant impact on infant cognitive development in LMIC settings. The importance of these findings are underlined by strong evidence from HICs for the continuity of cognitive ability from infancy to later life [93–95]. The consequences of early cognitive impairments in LMIC settings might be even more wide reaching, perpetuating a cyclical pattern of lost potential and generational poverty [6]. Despite an increase in the number of calls for the prioritisation of perinatal mental health in LMIC settings [96], it is an area that has typically been neglected in these contexts [97]. This review adds to the call for improved perinatal mental health services that will provide mutual benefits to the mother and the child. Research studies in this review demonstrated low adherence to the requirements for testing the independent effects of maternal depression during the antenatal and postnatal periods. Given evidence from HICs that antenatal and postnatal depression exert independent effects on cognitive development, it is important that future studies are designed in a way that allows researchers to isolate the effects of each. This can be achieved in two ways. Firstly, studies which have explored trajectories of maternal depression or split the sample into groups based on episode timing across the perinatal period should be replicated with the purpose of isolating independent and cumulative effects of perinatal depression [54, 66, 69]. Alternatively, a study can control for the effects of maternal depression at other relevant timepoints, as demonstrated by Tran et al. [55].

Given that the Bayley-III was designed to minimise previous cultural bias, the observation that significant impairments were more consistently found when using the Bayley-III as compared to the Bayley-II suggest that effects may be sensitive to the choice of instrument. This thought is reinforced by the observation that the two postnatal studies which used outcome measures designed specifically for their population also found significant effects of depression [58, 68]. Therefore, an important step in developing the LMIC literature will be the selection or development of culturally sensitive measures of cognitive development. In addition, due to the existing reliance on screening tools, where possible future studies should incorporate designs which facilitate comparisons of the effects of the mild-moderate and more severe, clinical levels of depressive symptoms.

Furthermore, while this review highlights the need for more high-quality studies investigating main effects, future studies should also look to explore a more complex relationship between perinatal depression and cognitive outcomes. In the light of recent advances in developmental science, investigations of a simple relationship between the two factors, even when controlling for a number of other variables, are inadequate [9]. While it is clear that the level of risk is generally higher in LMIC settings, it is not clear how the relationship between risk and perinatal depression functions [66]. Rather than simply controlling for possible confounding variables, studies need to investigate how variables such as infant temperament, parenting quality, infant gender and other key socio-economic indicators influence the relationship between perinatal depression and cognitive development [21, 98].

More specifically, future studies need to be designed to answer questions arising from current theories and hypotheses in developmental psychology. A number of theories, such as the foetal programming hypothesis [99], the predictive adaptive response hypothesis [100] and the differential susceptibility hypothesis [101] are increasingly influencing research in HICs and it is important for research LMICs to be similarly driven by current theory. The need for this is particularly evident in investigations of the role of postnatal depression, where findings for a direct association are very inconsistent. There is now considerable evidence that depression can compromise a caregiver's ability to respond sensitively and contingently to a child, and that a child's temperament can, in turn, influence the quality of the caregiver-child relationship

and affect how susceptible they are to both positive and negative caregiving practices [22, 91]. Given the interactional, dynamic nature of the factors described above, it is apparent that testing only a simple, direct relationship could result in a significant underestimation of the effects of postnatal depression on child outcomes.

## 5. Conclusion

While based on a limited number of studies (n = 16), this systematic review gives valuable insight into what is known about the relationship between perinatal depression and infant cognitive development in LMIC settings. A key strength of this study is the focus on, and in-depth exploration of, the association between cognitive development and perinatal depression alone. This contrasts with previous systematic reviews which have focused more broadly on a range of mental health disorders and neurodevelopmental outcomes, thereby retaining a high degree of novelty and adding considerable value to the existing literature. The findings provide a platform for the progression of research in this area by highlighting methodological and theoretical improvements that can be made to increase its relevance and impact.

Evidence for a direct association between antenatal depression and cognitive development was quite consistent, especially when restricted to prospective studies. There was some evidence that this association is independent of postnatal depression and endures into the child's second and third year. This provides a strong case for the inclusion of screening for maternal depression as part of routine antenatal care and for early intervention to be implemented in LMIC settings. Evidence for the direct influence of postnatal depression is more equivocal at present.

Importantly, findings from HICs suggest that the inconsistent findings regarding postnatal depression may be the result of its effect being evident in interaction with other factors. While there is some evidence of possible moderated effects of postnatal depression on cognitive development in LMICs, the majority of studies reviewed did not investigate these underlying pathways. The paucity of studies exploring anything more than a direct relationship between both postnatal and antenatal depression and cognitive development precludes drawing firm conclusions about this. More high-quality studies are needed in LMIC settings, driven by current theory, that test direct associations and examine moderating or mediating pathways to child cognitive development.

## Supporting information

**S1 File. Review protocol.**
(PDF)

**S2 File. PRISMA checklist.**
(DOC)

**S1 Table. Antenatal main effects.**
(DOCX)

**S2 Table. Postnatal main effects.**
(DOCX)

## Author Contributions

**Conceptualization:** Matthew Bluett-Duncan, M. Thomas Kishore, Veena A. Satyanarayana, Helen Sharp.

**Data curation:** Matthew Bluett-Duncan, Divya M. Patil.

**Formal analysis:** Matthew Bluett-Duncan.

**Investigation:** Matthew Bluett-Duncan, Divya M. Patil.

**Methodology:** Matthew Bluett-Duncan, Divya M. Patil, Helen Sharp.

**Project administration:** Matthew Bluett-Duncan.

**Supervision:** M. Thomas Kishore, Veena A. Satyanarayana, Helen Sharp.

**Writing – original draft:** Matthew Bluett-Duncan.

**Writing – review & editing:** Matthew Bluett-Duncan, M. Thomas Kishore, Veena A. Satyanarayana, Helen Sharp.

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
