## [Decision Letter · Decision Letter 0]

19 Mar 2021

PONE-D-21-04110

A systematic review of the association between perinatal depression and cognitive development in infancy in low and middle-income countries

PLOS ONE

Dear Dr. Bluett-Duncan,

Thank you for submitting your manuscript to PLOS ONE. After careful consideration, we feel that it has merit but does not fully meet PLOS ONE’s publication criteria as it currently stands. Therefore, we invite you to submit a revised version of the manuscript that addresses the points raised during the review process.

We look forward to receiving your revised manuscript.

Kind regards,

Angela Lupattelli, PhD

Academic Editor

PLOS ONE

Journal Requirements:

3. Please ensure that you refer to Figure 2 in your text as, if accepted, production will need this reference to link the reader to the figure.

Reviewers' comments:

Reviewer's Responses to Questions

**Comments to the Author**

1. Is the manuscript technically sound, and do the data support the conclusions?

Reviewer #1: Yes

Reviewer #2: Yes

2. Has the statistical analysis been performed appropriately and rigorously? 

Reviewer #1: Yes

Reviewer #2: N/A

3. Have the authors made all data underlying the findings in their manuscript fully available?

Reviewer #1: Yes

Reviewer #2: No

4. Is the manuscript presented in an intelligible fashion and written in standard English?

Reviewer #1: Yes

Reviewer #2: Yes

5. Review Comments to the Author

Reviewer #1: The question of whether or not maternal depression during the antenatal or postnatal periods influences cognitive development from birth to 3-years old in low- and middle-income countries is an important research question. This meta-analyses included 8 antenatal studies, 4 from South Africa, one from the Ukraine, one from China, one from Mexico and one from Vietnam. Postnatal studies were included from 8 countries, Pakistan, Bangladesh, Uganda, Barbados, South Africa, India, Brazil and Vietnam.

The study from Vietnam found that antenatal depression was associated with lower cognitive scores at 6 months after controlling for postpartum and concurrent depression symptoms. In China, maternal depression was associated with lower language scores on the Gesell Scale at 24 - 30 months.

Three other studies found a relationship between antenatal depression on cognitive outcomes but did not control for postnatal or concurrent maternal depression.

Authors reviewed 9 studies of the relationship of postnatal depression on infant cognition at different time points between birth and 36 months, depending on the study. Studies took place in Pakistan, Bangladesh, Uganda, Barbados, South Africa, India, Brazil and Vietnam. Of these studies, 2 were assessed as poor quality, 6 as fair quality and one as good quality. Four of the studies found a main effect of postnatal depression resulting in lower scores on infant language development.

The study also reviewed evidence for an effect of postnatal depression on infant development. In the settings of LMIC, these studies did not find an effect of postnatal depression on infant language development.

What does this study add?

Maternal depression during pregnancy adversely influences infant cognitive development. However, postnatal depression did not influence infant cognitive development in these settings. This finding raises questions about the role of environment in low and middle income countries on infant development and on the mother.

The reference list is comprehensive.

The focus on low and middle-income countries is important.

Reviewer #2: Review of --“ A systematic review of the association between perinatal depression and cognitive development in infancy in low and middle-income countries”.

By Matthew Bluett-Duncan, M. Thomas Kishore, Divya M. Patil, Veena A. Satyanarayana and Helen Sharp

The impact of perinatal depression on cognitive outcome of offspring is an important area that still blurred and requires more insights for the future-direction of child-focused interventions. The authors made a good attempt to review the existing assorted literatures on this complex topic and summarize the take-home messages.

The authors very clearly and elaborately described all the steps of the systematic review with references. They also detailed the working definitions of both exposure and outcome variables.

However, they are encouraged to address the following concerns:

•For simplifying the findings, the authors made lots of sub-headings in result and discussion section that caused repetition of similar information This repetition is distorted the flow of reading due to lack of customization. So, some sections could be merged, e.g. “key adjustment sections” does not need to be separately reported. Rather “adjustment information” can be blend with specific studies--- but the tables( 3 & 4) can be presented for better understanding.

• Table 1 & 2 on antenatal & postnatal findings, need to be more informative. At least those cab include positive/negative findings. Otherwise it is difficult to relate results and discussion on antenatal or postnatal findings with several tables in main paper and in supplementary document.

•Sample size of study population and their follow-up percentages need careful checking. For example, sample size and follow up % of the study done by Garman et al in Table-2 is not matching with the journal information. In the journal, at follow-up, 58% children were assessed for developmental measures. Please clarify if the sample size in the tables refers to the numbers of mother-child dyads.

•Reference numbers are wrong in many places e.g. page 17, section 3.3.1, 2nd line, reference 56 will be 58; page 19, ref of Rotheram-Fuller will be 55 etc.

• Throughout the write up please correct “BSID-III”-- It should be Bayley-III.

• Finally based on the nature of majority of the studies (mostly prospective or observational and sometimes 2ndary analysis), heterogeneous nature of the exposure/outcome assessments and huge drop-outs at follow up-- it will be better to say “significant association” of antenatal or postnatal depression with cognition, rather than “significant effect”, which is a more robust terminology.

6. PLOS authors have the option to publish the peer review history of their article (what does this mean?). If published, this will include your full peer review and any attached files.

Reviewer #1: No

Reviewer #2: No

---

## [Author Response · Author response to Decision Letter 0]

8 Jun 2021

Reviewer 1 comments to the author: no queries raised.

Reviewer 2 Comment: For simplifying the findings, the authors made lots of sub-headings in result and discussion section that caused repetition of similar information This repetition is distorted the flow of reading due to lack of customization. So, some sections could be merged, e.g. “key adjustment sections” does not need to be separately reported. Rather “adjustment information” can be blend with specific studies--- but the tables( 3 & 4) can be presented for better understanding.

Author Response: The “key adjustment” section was originally included as a distinct section as one of the key questions being asked by the review pertains to the ability of each study to isolate the variance in cognitive development that could be attributed to perinatal depression. It was felt that including this as a separate section was therefore justified. However, it is a valid point that its inclusion as a distinct section may not be necessary due to the inclusion of the information elsewhere, and that it serves to break up the flow of the narrative. Following the advice of the reviewer, this section has now been removed but the tables have been retained. The tables are now referenced in the initial descriptions of the antenatal and postnatal studies and the relevance of the information regarding the “key adjustment variables” is referred to and discussed in the discussion section. We believe this retains the emphasis on the importance of these factors but removes some of the repetition. 

 We also considered whether any of the other sub-sections could be reduced or merged together, particularly in the “Results” section. However, we believe that the way they are currently set is optimal as it allows the reader to clearly distinguish between antenatal (section 3.2) and postnatal studies (section 3.3), and within them, those studies which found significant associations and those that did not. Similarly the layout of the discussion section allows the reader to clearly distinguish between the synthesis of the antenatal studies and the synthesis of the postnatal studies. We also believe that a separate section for methodological considerations is justified due to the emphasis in the review on assessing the robustness of the studies carried out in LMIC setting and the importance of clearly evaluating how different methodological decisions may have impacted results. While it may be possible to merge the discussion regarding methodological issues with the discussion of overall findings, the result would be a loss of clarity for both areas.

Reviewer 2 Comment: Table 1 & 2 on antenatal & postnatal findings, need to be more informative. At least those cab include positive/negative findings. Otherwise it is difficult to relate results and discussion on antenatal or postnatal findings with several tables in main paper and in supplementary document.

Author Response: We acknowledge that it would be beneficial to the reader to be able to easily see from the tables whether a study found a significant association or not. Therefore, we have now added a column to tables 1 & 2 to indicate whether or not a significant association has been found. We have not added any further information to these tables as the results section already describes the key findings from each study and we did not want to add in overly repetitive information. The two supplementary tables show a detailed breakdown of the study results for the antenatal and postnatal periods. 

Reviewer 2 Comment: Sample size of study population and their follow-up percentages need careful checking. For example, sample size and follow up % of the study done by Garman et al in Table-2 is not matching with the journal information. In the journal, at follow-up, 58% children were assessed for developmental measures. Please clarify if the sample size in the tables refers to the numbers of mother-child dyads.

Author response: The reviewer is correct that the follow-up rate for Garman et al was incorrect on the table due to a miscalculation by the lead author. This has now been rectified and all the other follow-up rates checked carefully, with a couple of further minor revisions being made. Follow-up rates were calculated from the information provided by each study regarding the number of mother-infant dyads who completed developmental assessments and/or were included in the final analyses. This has now been clarified in the text as well.

Reviewer 2 Comments: Reference numbers are wrong in many places e.g. page 17, section 3.3.1, 2nd line, reference 56 will be 58; page 19, ref of Rotheram-Fuller will be 55 etc.

Author Response: The reviewer was correct in noticing that references were incorrectly labelled in a few instances. References have now been checked and corrected where necessary.

Reviewer 2 Comments: Throughout the write up please correct “BSID-III”-- It should be Bayley-III

Author Response: This has now been changed to Bayley-III. BSID-II has also been changed to Bayley II for consistency.

Reviewer 2 Comments: Finally based on the nature of majority of the studies (mostly prospective or observational and sometimes 2ndary analysis), heterogeneous nature of the exposure/outcome assessments and huge drop-outs at follow up-- it will be better to say “significant association” of antenatal or postnatal depression with cognition, rather than “significant effect”, which is a more robust terminology.

Author Response: This is a very good point and we have now changed the majority of references to “significant effects” to “significant associations”. There are a few instances where the use of the word “effect” remains more suitable and so it has been retained. Importantly, key sentences that serve to summarise findings in the abstract, discussion and conclusion sections have now been changed to reflect the nature of the studies that were synthesized.

---

## [Editor Report · Decision Letter 1]

14 Jun 2021

A systematic review of the association between perinatal depression and cognitive development in infancy in low and middle-income countries

PONE-D-21-04110R1

Dear Dr. Bluett-Duncan,

We’re pleased to inform you that your manuscript has been judged scientifically suitable for publication and will be formally accepted for publication once it meets all outstanding technical requirements.

Kind regards,

Angela Lupattelli, PhD

Academic Editor

PLOS ONE

---

## [Editor Report · Acceptance letter]

18 Jun 2021

PONE-D-21-04110R1 

A systematic review of the association between perinatal depression and cognitive development in infancy in low and middle-income countries 

Dear Dr. Bluett-Duncan:

I'm pleased to inform you that your manuscript has been deemed suitable for publication in PLOS ONE. Congratulations! Your manuscript is now with our production department. 

Kind regards, 

on behalf of

Dr. Angela Lupattelli 

Academic Editor

PLOS ONE